# Where can rewetting of forested peatland reduce extreme flows? – model experiment on the hydrology of Sweden

Maria Elenius[1*], Charlotta Pers[1], Sara Schützer[1], Göran Lindström[1], Berit Arheimer[1]

[1]Hydrological Research Unit, SMHI, Norrköping, 601 76, Sweden

[*]Present address FOI, Linköping, 583 30, Sweden

*Correspondence to*: Maria Elenius (maria.elenius@foi.se)

**Abstract.** Historical drainage to improve forestry practices has resulted in 0.6-0.7 million hectares drained forested peatland in Sweden. This has reduced the storage of water in the landscape and may impact greenhouse gas emissions, biodiversity and the damping of extreme water flows. National restoration actions therefore aim at rewetting 0.1 million hectares of forested

peatland in Sweden, despite the limited and sometimes contradictory evidence in the impacts of rewetting. To clarify the potential impact on extreme flows and their cause-effects relationships from rewetting, we simulated flow under various conditions of the climate, local hydrology and rewetting practices (ditch blocking alone or combined with reduced tree cover). For this, we used the HYPE model setup across Sweden (450 000 km$^2$) with improved calculations of runoff in drained forest and routines for recharge and discharge areas. National evaluation of changes in discharge extremes was combined with a

detailed study in south-east Sweden, with the aim to understand rewetting impacts at various scales. We found that the change in discharge extremes from catchments of 10 km$^2$ is small, because there is considerable mixing with runoff from various landcover. Hence, at the larger scale, rewetting is not an efficient measure to combat droughts or floods. However, for ecosystems in the streams only draining peatlands, rewetting can have an impact if appropriate sites for restoration are selected. The results show that groundwater level prior to rewetting and reduced tree cover are governing the effect on water runoff.

Wetland allocation and management practices are thus crucial if the purpose is to reduce flow extremes in peatland streams.

## 1 Introduction

There is clear evidence in existing scientific literature that the climate is changing (IPCC, 2007) in a way which goes beyond our present experience and exceeds our preparedness, e.g. adaptation to water risks (Sörensen and Mobini, 2017). Changes of extremes, such as hot/cold days, warm-spell duration and heavy rainfall, all affect the hydrological cycle and thereby the water

security. We see these effects also in cold-temperate and subarctic climates, for instance, the dry period in Sweden during spring and summer 2018 was attributed to climate change (Vogel et al., 2019) and led to water scarcity, with severe problems for agriculture, and forest fires. An increase in temperature leads to an exponential increase in the moisture holding capacity of the air with increasing precipitation noted over Fennoscandia (Westra et al., 2013) as well as record-breaking daily precipitation extremes (Lehmann et al., 2015). Cloudbursts and flooding in Sweden are reported to trigger enhanced transport

of chemical and microbial pollutants, e.g. from sewer overflow (Olsson et al., 2013) as well as erosion and landslides, nutrient transport (Wu and Malmström, 2015) and acidification (Erlandsson et al., 2010). Hence, water in Sweden follow the global tendencies in becoming too little, too much, and more polluted as an effect of global warming. Being top-5 on the list of countries with most lakes in the world (Messager et al., 2016), Sweden has profited from lakes dampening high-flows and buffering against low flow. However, climate change may put new stress on water management and urgent actions are thus needed due to signs of enhanced competing interests for sustainable water-related security.

One method that has been proposed to further dampen high flows and buffer against low flow is the rewetting of historically drained forested peatlands. The aspiration has been to return to undrained conditions of peatlands acting as sponges, storing rainfall during storms and then gradually releasing the water in dryer periods (Holden, 2005). It has been estimated that as much as 87 % of wetlands globally may have been degraded by human activity since 1700 (Davidson, 2014) and Sweden follows this trend; Holmen (1964) estimated that around 0.7 million hectares forested peatland was drained only in Sweden in the period 1873 - 1960. Rewetting of peatlands by ditch blocking is a policy action not the least in this country where a long-term goal is to rewet 0.1 million hectares of drained forested peatlands (Drott and Eriksson, 2021), currently with a focus on greenhouse-gas reductions, but also considering other ecosystem services such as damping flow fluctuations.

Previous studies, largely based on evidence from Finland, Canada and the UK, show variable ability of both natural (or unaltered, undrained) and rewet peatlands in terms of damping flow extremes. The damping in unaltered peatlands has been found to depend on the antecedent storage prior to rainfall events (Acreman and Holden, 2013), the position in the landscape (Åhlén et al., 2022), and flow path structure and catchment size (Edokpa et al., 2022). Karimi *et al.* (2023) found no significant attenuation of floods from peatlands in a recent study involving 9 years of hydrometric data from 14 catchments in northern Sweden, and claimed that this could be due to the overshadowing impact of other land cover types in the catchment. Arheimer and Pers (2017) showed that previous efforts with constructing 1574 wetlands in the southern half of Sweden, for damping flows to allow nutrient removal, had very minor effects on nutrient transport from land to sea. The wetland area remained very minor and the constructions were not done in optimal locations for nutrient retention.

In terms of the impact of wetland restoration measures such as rewetting peatlands, a recent review on temperate and Boreal forests by Bring et al. (2022), showed that groundwater levels 1 m from the intervention increased on average by 0.45 m but the effect was reduced by a factor of two already at 9 m distance. This was compared with drainage which had a similar change in near-ditch groundwater levels from undisturbed conditions by -0.42 m but here the effect was reduced to 50 % at a larger distance of 21 m, meaning that it may be difficult to reverse drainage impacts away from ditches. The authors tried to relate the variable restoration impact to peat depth, time since intervention, intervention magnitude, soil type, ditch spacing, transect type and climate zone, but no significant results were obtained, except that restoration of blanket bogs (not included in numbers above) had small effects on groundwater levels.

In contrast, Holden et al. (2011) found no discernable effect on groundwater and Karimi et al. (2024) found only 0.03 m increase in the groundwater level in addition to the change of a reference site during the same period. The literature also shows variable

impacts on runoff and discharge after peatland restoration, e.g. with peak flows reduced (Menberu et al., 2018; Wilson et al., 2010) or with peak flows sometimes reduced, sometimes increased (Ballard et al., 2012). The large knowledge gaps in the fundamental drivers of the hydrological response from rewetting forested peatland, poses challenges in how to allocate societies' resources effectively, when rewetting is applied to improve water security. Important efforts to acquire more data on peatland hydrology before and after rewetting in various settings are currently being pursued but are costly and time consuming. A low-hanging fruit that can provide direct insights as well as guide data collection is the analysis of hydrological simulation results with variable inputs. Previous simulation studies on selected catchments in a Swedish context show small impacts from changed ditch drainage on both low flows (Lindström, 2019; Stensen et al., 2019) and high flows (Johansson, 1993). Complementary large-scale simulations are needed to better discern the impacts and the driving factors of the variable results from rewetting reported in the literature.

Here, we make use of a hydrological model applied at the national scale (Strömqvist et al., 2012) that explicitly simulates groundwater levels, runoff and discharge for entire Sweden (450 000 km$^2$), represented by approximately 40 000 sub-catchments, each with up to 116 hydrological response units (HRUs) to account for spatial variability (see Appendix A). The aim of this study is to understand the main drivers behind the heterogeneous impacts of rewetting on discharge extremes. We draw on a recently published national dataset of ditches (Lidberg et al., 2023) and we present simulated rewetting impacts on discharge (i.e. the accumulated discharge from upstream areas and the accumulated local runoff) as well as the local impacts on peatland groundwater levels and peatland runoff. More importantly, we carefully examine how these impacts depend on peatland properties, the drained state, the position in the landscape, and the type of rewetting performed. This sensitivity study focuses on the 882 sub-catchments in the Motala ström catchment in south east Sweden, which have large variability in land use, soil types and precipitation. We distinguish between two important aspects of rewetting – the direct impact due to removal of hydrological pathways (ditches), and the increased soil wetness following reduced tree density, which can either be the result of tree removal or of trees being unable to cope with a wetter environment after ditch removal.

## 2 Methods and data

Here we describe the study areas and the hydrological model, focusing on aspects important for rewetting impacts, and the sensitivity matrix that was applied to draw conclusions on the important drivers of varying rewetting performance. Throughout this text, we will use the term "forested peatland" for forests on peat and fens in the landscape.

### 2.1 Study areas

We study the entire country of Sweden, situated in Northern Europe, and perform a more detailed study of the Motala ström catchment in south east Sweden, see Fig. 1. Sweden was previously glaciated and has subarctic and cold-temperate climates,

with Motala ström catchment in the cold-temperate climate zone. The land use and peat cover presented below are based on national data sets introduced in Section 2.2.

Sweden is largely covered by forest, making up 61 % of the total surface area, whereas peat soils cover 17 % of the entire
surface and cover 7 % of forests. Assuming that drained conditions extend 20 m laterally from ditches, 1.4 % of the forested area in Sweden constitutes drained forested peat, or 630 000 ha, which is not far from the 650 000 – 700 000 ha estimated by Holmen (1964) for the period 1873-1960. Holmen assumed 6 ha drained conditions per km length of ditch, which corresponds to 30 m to either side of the ditch, if ditches do not intersect within this distance.

The Motala ström catchment (1.5 M ha) covers both forested and cultivated areas, and also peatland. The distribution is similar
to the national scale with 53 % forest, 9 % peat, 7 % forested peat and 0.3 % of sub-catchments having more than 10 % drained forested peat (1 % have more than 5 % and 44 % have more than 1 % drained forested peat). The large Lake Vättern in the western part of the catchment (gray in the figure) is the second largest lake in Sweden and sixth-largest lake in Europe. It drains to the Motala ström River (also marked in gray in the lower panels) with an outlet in the Baltic Sea. There are many lakes in this catchment (as in Sweden in general), and several of them, including Vättern, are regulated for hydropower.

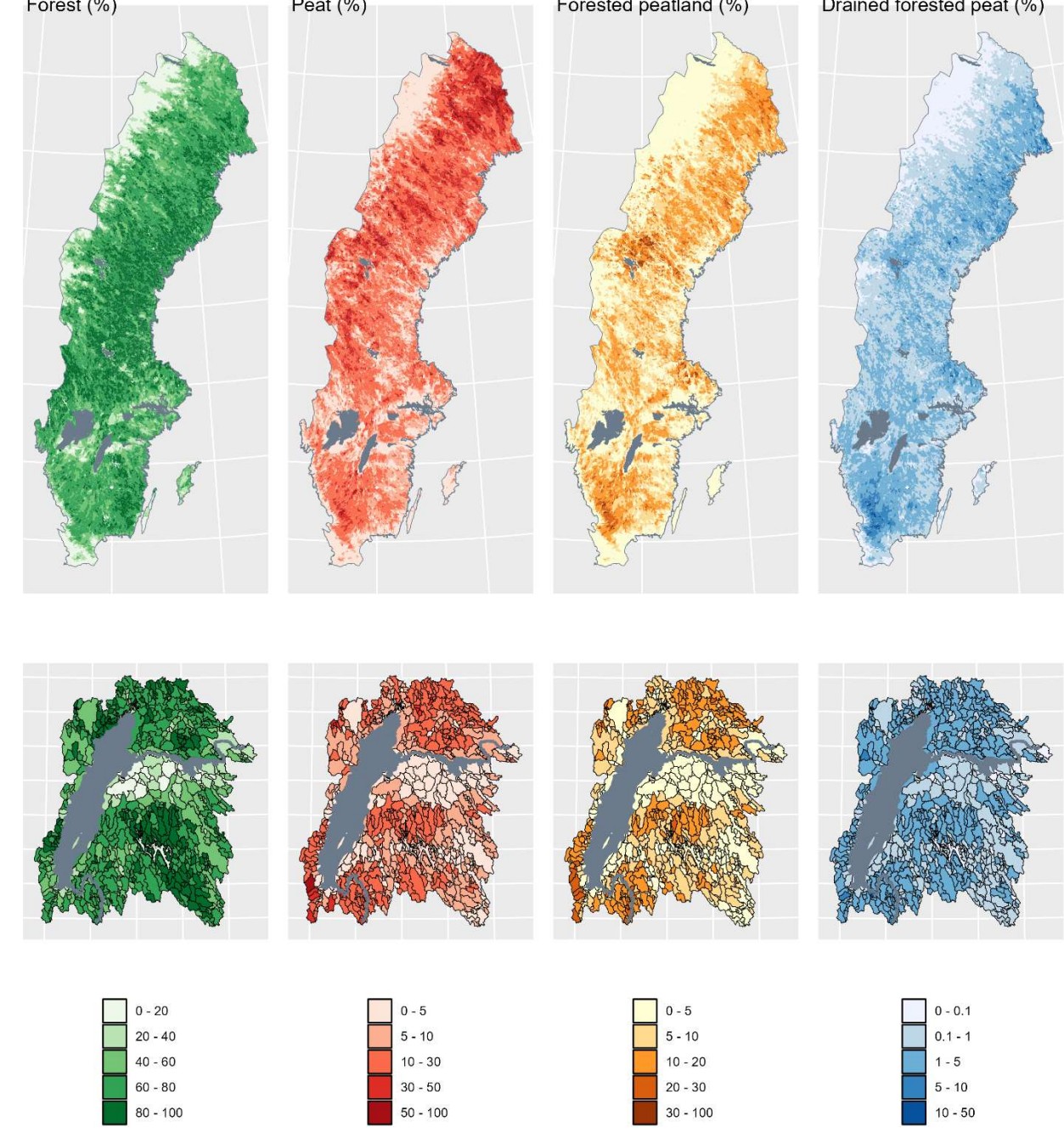

**Figure 1: Peatland and drainage in Sweden, including a more detailed view over the Motala ström catchment.**


## 2.2 Hydrological data and sensitivity matrix

### 2.2.1 Simulation setup

The impact of rewetting was assessed based on simulations of drained and rewet conditions using the hydrological model HYPE (Lindström et al. 2010) as set up for Sweden, S-HYPE (Strömqvist et al. 2012 and 2022; version 2016i). This model version has a very good description of Swedish daily water flow (average NSE 0.86 and volume error - -3 % for 157 river gauges over the calibration period 2006 – 2020, or 0.74 and -2 % for 350 gauges when including stations affected by hydropower). The model is used operationally for national monitoring with indicators on hydrological status of

rivers/waterbodies, design values for buildings and infrastructure, climate change impact assessments and the national service of flood warnings and notifications of low flow. S-HYPE is calibrated based on HRUs representing mainly combinations of land use and soil type (Appendix B), which enables assessment also of ungauged basins (Arheimer and Lindström, 2013). In the current study, evaluation is based on simulations covering the 10-year period between 2012 and 2021, following a 5-year initialization period (2007-2011). The model uses precipitation and temperature as forcing, and results are provided at a

daily time step for sub-catchments with an average size of 10 km$^2$ (or 1 000 ha; deriving ~40 000 sub-catchments for Sweden and 882 for Motala ström). Within each sub-catchment, calculations for land use and soil classes are performed in up to 116 HRUs.

In the current model setup, land cover and soil data has been collected from various sources (Strömqvist et al., 2022). Forest land-use data for different kinds of forest, including forested wetland, was obtained from the national landcover data "NMD"

(Swedish Environmental Protection Agency, 2023) with 10 m resolution. The forested wetland was here combined from NMD-classes 1.2-1.7 which have "tree-covered areas on wetlands with a total crown cover of > 10 %". We interpret this as fens (Swedish: *kärr*) since other wetlands less frequently have forest. Soil-type data (including peat) of forested land, with varying resolution between 1:25 000 to 1:750 000, was collected from the Swedish Geological Survey (2024). Total soil depth was collected and simplified from Daniels and Thunholm (2014) and calculations are typically performed within three soil layers

that extend to depths originally motivated by top soils (affected by plough depth in case of agricultural fields), the remainder of the root zone, and any soil beneath the root zone. The main focus here is on forested peatlands, and the three soil layers of these HRUs extend to 0.25 m, 0.75 m and 1.5 or 2.25 m below the soil surface.

Appendix A provides an overview of the HYPE model and Appendix B explains the calibration method used in S-HYPE and further illustrates model performance and uncertainties related to the current study. Full model documentation and open source

code is found at https://hypeweb.smhi.se/model-water/, while the text below focuses on the aspects most relevant for the model experiment on rewetting peatlands.

### 2.2.2 Runoff description

Since our objective is to study the hydrological response to rewetting, the model description of runoff is essential. In HYPE, runoff from the HRUs occurs through the soils, in drainage tiles/ditches, and as surface runoff (which in Sweden mainly

consists of reels and temporary creeks) if the soil is saturated or the infiltration capacity is exceeded. Saturated surface runoff is calculated as a land-use dependent fraction *srrcs* of the free water above the soil surface that drains every day (Table 1). Runoff through soil depends on the soil saturation above field capacity, the groundwater table (pressure head) in relation to stream depth, and on a soil-type dependent recession coefficient *rccs* which is given as input for the top and bottom layers and calculated for the middle soil layer to fit an exponential decrease with depth. Drainage through ditches (or tiles, which have the same model description) occurs when the groundwater level is above the level of ditches, in which case a soil-type dependent fraction *trrcs* of the water at saturation above field capacity is drained every day. Only water above the depth of ditches is affected by this drainage, and it is also possible to limit the lateral extent of the impact as a fraction of the HRU area. In this case, *trrcs* is multiplied by this fraction to obtain a proportionally smaller runoff coefficient for ditches. In the description of runoff in the model, it is also possible to use a regionally calibrated correction factor *rrcscorr* such that *rrcs* = *rrcs* * (1 + *rrcscorr*) in each soil layer (Lindström 2016) and the same correction is used for *srrcs* and *trrcs*. There are 330 geographical parameter regions in the Swedish model setup with varying *rrcscorr*. We performed simulations both with and without this correction.

**Table 2: Runoff coefficients (unit day$^{-1}$)**

| Type of runoff | Runoff coefficient | Value without regional calibration | Regional calibration factor *rccscorr*, range in Sweden |
|---|---|---|---|
| Surface, wetlands | *srrcs* | 0.282 | -0.88 to 6.1 |
| Surface, forest | *srrcs* | 0.161 | -0.88 to 6.1 |
| Peat top layer | *rccs1* | 0.055 | -0.88 to 6.1 |
| Peat bottom layer | *rccs2* | 0.01365 | -0.88 to 6.1 |
| Ditches in peat | *trrcs* | 0.05 | -0.88 to 6.1 |

To describe the impact of ditch drainage, we first implemented new data on the location of Swedish ditches (Lidberg et al., 2023) in the hydrological model. The data does not contain information on the depth of ditches or the lateral impact of ditches. Piirainen et al. (2017) state that drainage is performed to 0.5-1.0 m depth in Swedish forests, and we chose 0.7 m as a baseline scenario. Based on the literature review of Bring et al. (2022), we defined the nearest 20 m of ditches as impacted by drainage. The model does not account for the gradual reduction of impact away from ditches, but assumes the ditches act directly on the region defined as drained. We analyzed what fraction of each forested HRU in each sub-catchment is located within 20 m from ditches ("drained"), and then grouped HRUs with similar drained fractions nationally, arriving at five groups of which two are considered here for rewetting, i.e. fens (4 % of Sweden's surface area) and other forested peatland (3 % of Sweden's surface area). Nationally, 22 % and 14 % of the soil in these groups is drained under our assumptions, but the simulations do account for the local percentage affected in each sub-catchment.

All runoff is directly routed to surface waters in the sub-catchment (first entering a generic "local stream"). As part of the current study, the possibility to first route runoff from recharge HRUs to discharge HRUs was developed, where a given percentage of the runoff from inflow HRUs enters the third soil layer of the outflow HRUs (e.g. peatlands) within each sub-catchment (see Appendix A). The idea behind this development was to be able to reflect differences related to the position in the landscape, which had been previously found to impact the hydrological response of peatlands (Åhlén et al. 2022). This also facilitates more accurate representation of peatlands typically occurring in topographic depressions with groundwater levels close to the surface (Bring et al., 2022).

### 2.2.3 Interception and evapotranspiration

Apart from the direct impact of rewetting on runoff through removal of hydrological pathways in ditches, other hydrological changes may also occur when forests are rewet. Most notably, tree density is often reduced, either by cutting trees or because conditions become too wet, and therefore interception and evapotranspiration are reduced (Lindström, 2019). This can be exemplified by the differences in calibrated S-HYPE parameters between existing forest wetlands and other forests, shown in Table 2, with less interception and evapotranspiration in wetlands. Challenges occur in distinguishing between the changes in interception and evapotranspiration in model calibration, but here the combined impact is considered.

Table 2: Example parameters that differ between forested wetland (>10 % crown cover) and other forest. The model also has calibrated values for open wetland (no tree cover) and these are the same as the "forested wetland" parameters in this table except for a slightly larger *cevp* in open wetland (0.087 mm $^{\circ}$C$^{-1}$ day$^{-1}$).

| Parameter | Unit | HYPE name | Forested wetland | Forest, not wetland |
|---|---|---|---|---|
| Removal fraction of precipitation due to interception | - | *pcluse* | 0.1 | 0.13 to 0.19 |
| Evapotranspiration parameter | mm $^{\circ}$C$^{-1}$ day$^{-1}$ | *cevp* | 0.079 | 0.135 to 0.155 |
| Threshold temperature for snow melt, snow density and evapotranspiration | $^{\circ}$C | *ttmp* | 0.39 | -0.29 to 0.0003 |

### 2.2.4 Sensitivity matrix and impact indicators

As discussed above (Section 1), there are large knowledge gaps in the fundamental drivers of variable rewetting success. To analyze this, a sensitivity study according to Table 3 was performed for the Motala ström catchment with respect to the most important factors, which we think are related to the efficiency of ditch drainage prior to rewetting ("influence" and "depth" in the table), peat properties ("regional calibration"), the position in the landscape ("recharge"), and the different aspects of

rewetting ("Rewet1" for removal of hydrological pathways and "Rewet2" where also losses to interception and evapotranspiration are reduced, and surface runoff is increased to represent wetlands). Starting from the baseline scenario ("A" in the table), the drainage efficiency of ditches in other scenarios was assumed to be the same or larger, with ditches affecting the full HRU and/or being twice as deep. Together with the assumptions of complete restoration to natural conditions, the sensitivity matrix is therefore likely producing overestimates of the impact in general, and this design was chosen because an

initial investigation (Schützer et al., 2023) showed insignificant impacts using the baseline scenario (A). Parameters describing the impact of ditches or recharge from other land were only changed from the baseline scenario for HRUs that would be rewet, whereas regional calibration of runoff (yes/no) was changed in all HRUs.

**Table 3: Sensitivity matrix. Recharge 30 % means that within sub-catchments up to 30 % of runoff from forest on other soil than peat is diverted to forested peatland, but the contributing area is at most a factor of three larger than that of the forested peatland.**
**Rewet 1 and 2 were tested on all cases A-H, where Rewet 1 is a removal of ditches only and Rewet 2 also changes the land use of the drained part (20 m or full) to forested wetland (cf. Table 2).**

| Drained | Influence (m) | Depth (m) | Recharge (%) | Regional cal. |
|---|---|---|---|---|
| A. baseline | 20 | 0.7 | 0 | Yes |
| B. all peat | Full | 0.7 | 0 | Yes |
| C. 1.4 m | 20 | 1.4 | 0 | Yes |
| D. 1.4 m, all peat | Full | 1.4 | 0 | Yes |
| E. recharge | 20 | 0.7 | 30 | Yes |
| F. recharge, all peat | Full | 0.7 | 30 | Yes |
| G. no regional calibration | 20 | 0.7 | 0 | No |
| H. no regional cal., all peat | Full | 0.7 | 0 | No |

The impact of rewetting was studied in terms of changes in the average yearly minimum and maximum groundwater level, runoff and discharge (positive values referring to increases with rewetting). Groundwater and runoff changes are expressed in

absolute terms (m and mm day$^{-1}$) to facilitate detailed analysis of the driving factors, but discharge is presented as percent change relative to the drained state. An exception is that the changes in minimum discharge is expressed as percent change relative to the drained *average* discharge rather than minimum discharge to avoid division by zero.

**3 Results and discussion**

Here we present the rewetting impacts, starting with discharge impacts at the national domain, which is followed by a
description of changes in discharge, peatland groundwater levels and peatland runoff in the Motala ström catchment. These
results found the basis for the analyses of the driving factors of the heterogeneity in rewetting impact.

**3.1 National rewetting impacts on discharge extremes**

Figure 2 shows changes in discharge extremes from the national evaluation of downstream impacts of rewetting using the
baseline conditions (case A, with Rewet1 and Rewet2, Table 3). The average of the minimum and maximum discharge per
year changed by less than 1 % in a vast majority of sub-catchments, and always less than 4 % with Rewet1. It changed less
than 5 % with Rewet2 except in a negligible number of sub-catchments (11 out of approximately 40 000, where the maximum
flow increased between 5 and 9 %). No sub-catchment with an upstream area larger than 44 km$^2$ had changes in minimum or
maximum discharge more than 1 % (the average upstream area of sub-catchments in the model is 630 km$^2$). We refer to
changes in discharge extremes less than 5 % as small in the sensitivity analysis of Motala ström below. This is of course
subjective, but can be compared with the assessment of ecological status according to the Swedish implementation of the
Water Framework Directive, where average daily volume changes less than 5 % do not invoke any reduction of status (Swedish
Agency for Marine and Water Management, 2019). The small impact at the scale of sub-catchments is related to the small
coverage of drained forested peatlands (Section 2.1) in relation to other combinations of land use and soil type. For example,
only 0.8 % of sub-catchments have more than 10 % drained forested peat, whereas 5 % have more than 5 % and 38 % have
more than 1 % drained forested peat.

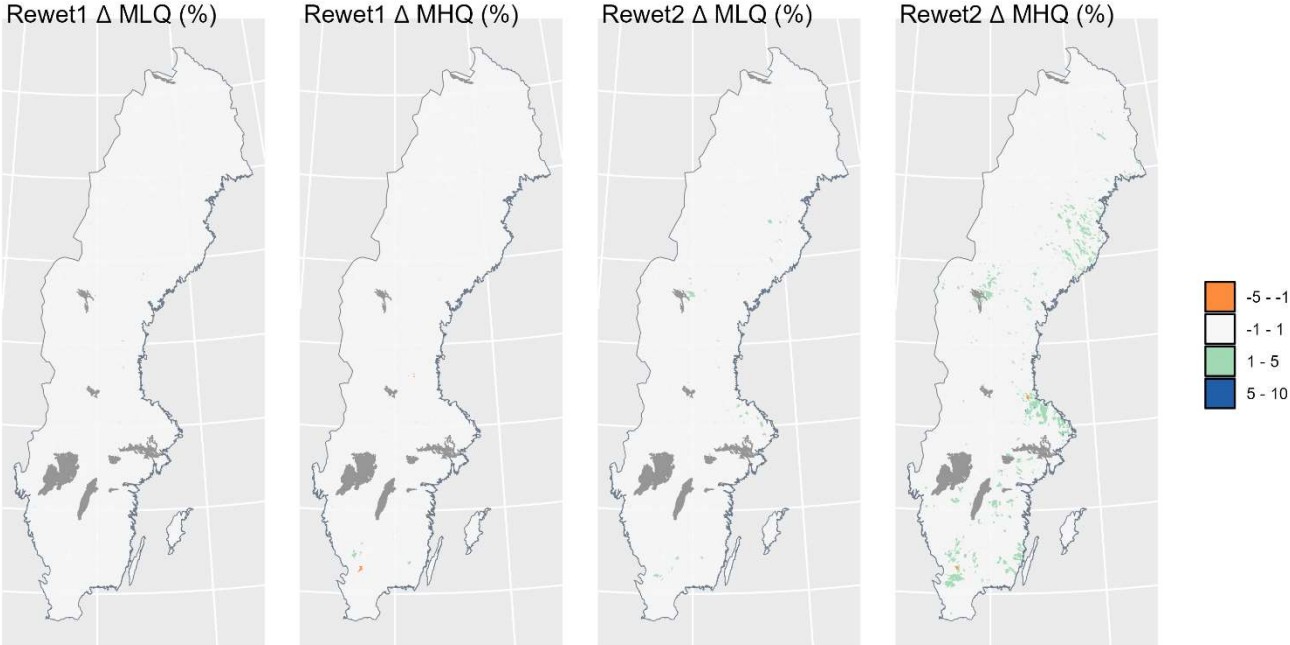

**Figure 2: Relative changes (%) in the average minimum (MLQ) and maximum (MHQ) yearly discharge, over the study period 2012-2021. Forested peatland and fens were rewet.**


## 3.2 Motala ström rewetting impacts

Following the national evaluation, we proceed with results from the Motala ström sensitivity study, and first analyze the impact of rewetting on discharge, see Fig. 3. All changes in discharge extremes are small under the most realistic assumption of 20 m influence of ditches (cases A, C, E, G on the left panel) regardless of the rewetting scenario (1 or 2), except in one instance.

With full influence of ditches (B, D, F, H on the right panel), changes to minimum discharge are also small (except in five sub-catchments), but here, there are substantial increases in maximum discharge in some sub-catchments, up to 22 %. This increased maximum discharge was found mostly in the central part of the catchment (Fig. 4). These results for the full lateral influence of ditches are estimated to be unlikely to occur, and are given as an estimate on the upper bound of possible impact. The impact of regional calibration on minimum and maximum discharge was found to be negligible.


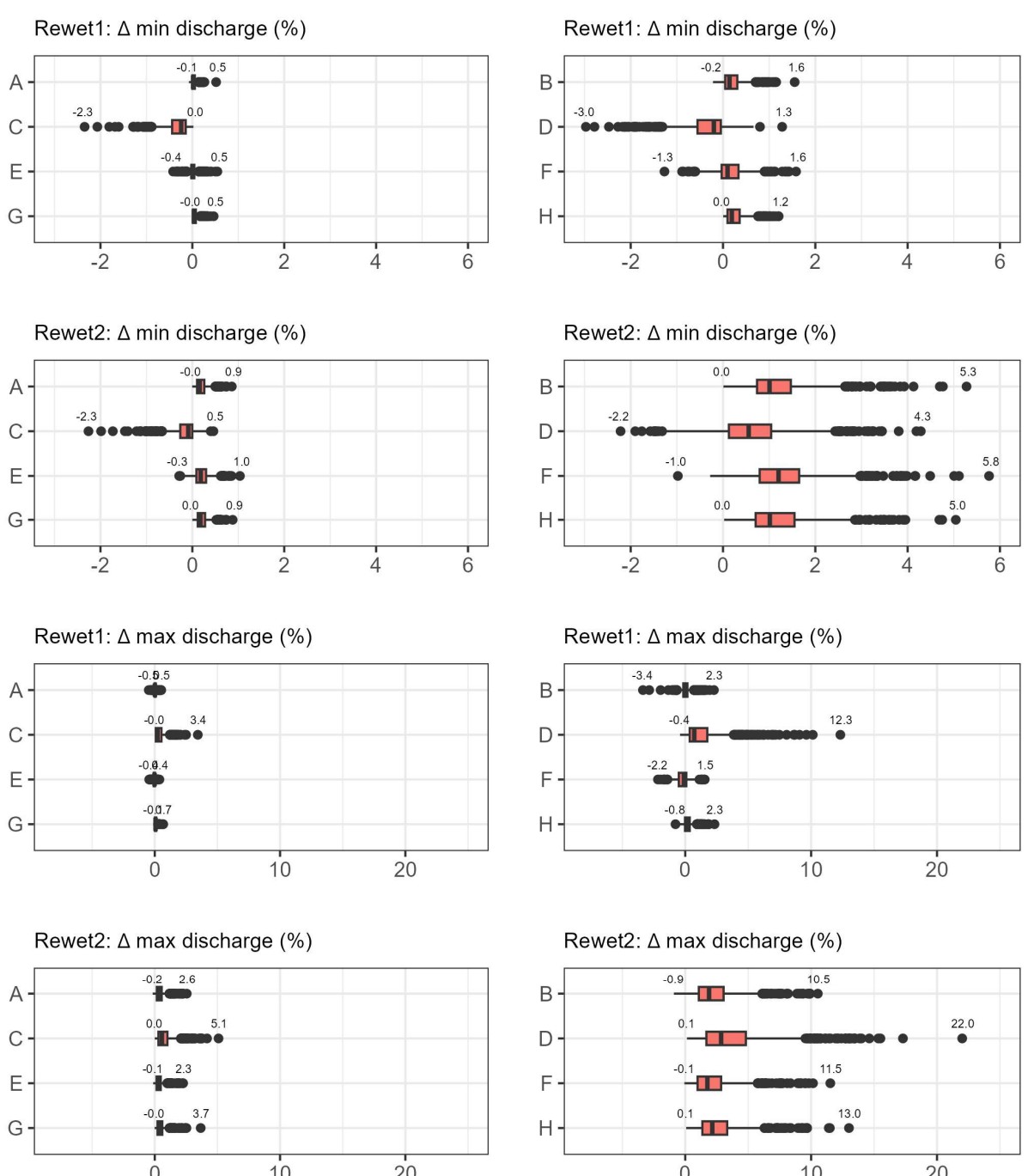

**Figure 3: Changes in sub-catchment average yearly minimum and maximum discharge with rewetting. The statistics are based on the 656 sub-catchments that have coniferous forest on peatland (depth 1.5 m). Cases with 20 m influence (A, C, E, G) are presented in the left panels and cases with full influence (B, D, F, H) in the right panels.**


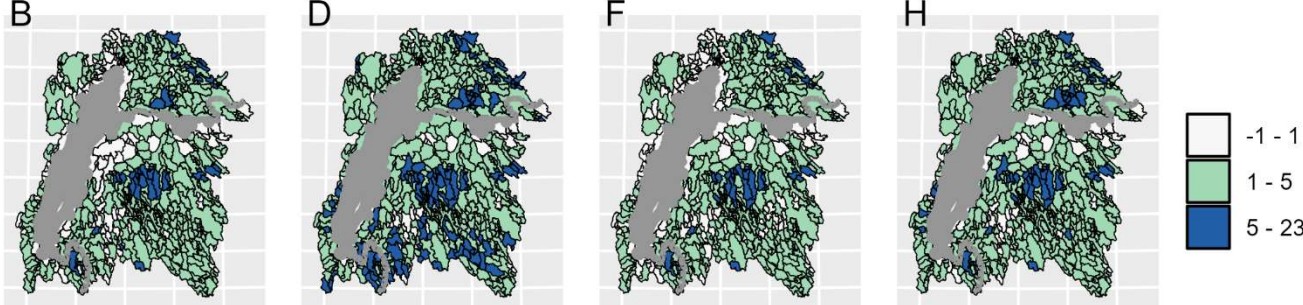

**Figure 4: Relative changes (%) in maximum yearly discharge after the Rewet2 scenario (ditch plugging and reduced tree cover) in**
**the extreme case of ditches impacting the full HRUs.**

The discharge extremes in the realistic case of 20 m lateral influence of ditches were therefore small at sub-catchment outlets, where runoff from rewet peatlands is mixed with other runoff, similar to the conclusions of (Johansson, 1993; Karimi et al., 2023; Lindström, 2019; Stensen et al., 2019). To understand if discharge extremes could be larger for small rivers mainly
draining rewet peatlands, i.e. when the discharge mainly represents runoff from the restored soils, we also analyzed the changes in peatland runoff extremes with rewetting. This varies by type of forest, and we chose to present results for coniferous forest with depth 1.5 m which is one of the most common forest types on peatland in Motala ström (8 400 ha), although for Rewet2, the land use was always changed to fens, which originally covered 67 000 ha in the catchment. (The model also has 800 ha coniferous forest on peat with 2.25 m depth.) Runoff extremes are closely linked with groundwater extremes which are
therefore also presented. For this analysis, the cases with full influence of ditches are described (B, D, F, H), to show local conditions in soil that is initially drained.

We begin with an examination of the runoff exceedance curves of the examined HRU (Fig. 5), to get a sense of the magnitudes of both low and high runoff. We refer to runoff exceeded 5 % of days as R05 and runoff exceeded 95 % of days as R95. The drained state ("Drained") has R95 in the range (5th to 95th percentile between sub-catchments) 0.006 to 0.03 mm day$^{-1}$ at
reference case B, with very similar values in case H (no regional calibration) and generally lower for deep ditches (D, 4e-5 to 0.03 mm day$^{-1}$ not fully shown in the figure) and higher with recharge (F). Rewet1 does not change R95 much except if ditches were deep (D, lower range increasing to 0.006 mm day$^{-1}$). Rewet2 generally gives much higher R95 (as desired), with the lower/upper limits of the range increasing by a factor 5/7 (B), 750/7 (D), 2/4 (F) and 10/3 (H). The large factor of increase for the lower range of D means that we do get runoff, i.e. it should not be used to generalize impacts of rewetting.

R05 at the drained reference case (B) varies in the range 0.8 to 2 mm day$^{-1}$ and is similar for other cases except with recharge (F) where the range is 3 to 8 mm day$^{-1}$. Results are again not much changed with Rewet1, whereas Rewet2 mostly gives increased R05 (unfortunately), by a factor 2 to 3, except in F which has small changes in R05.

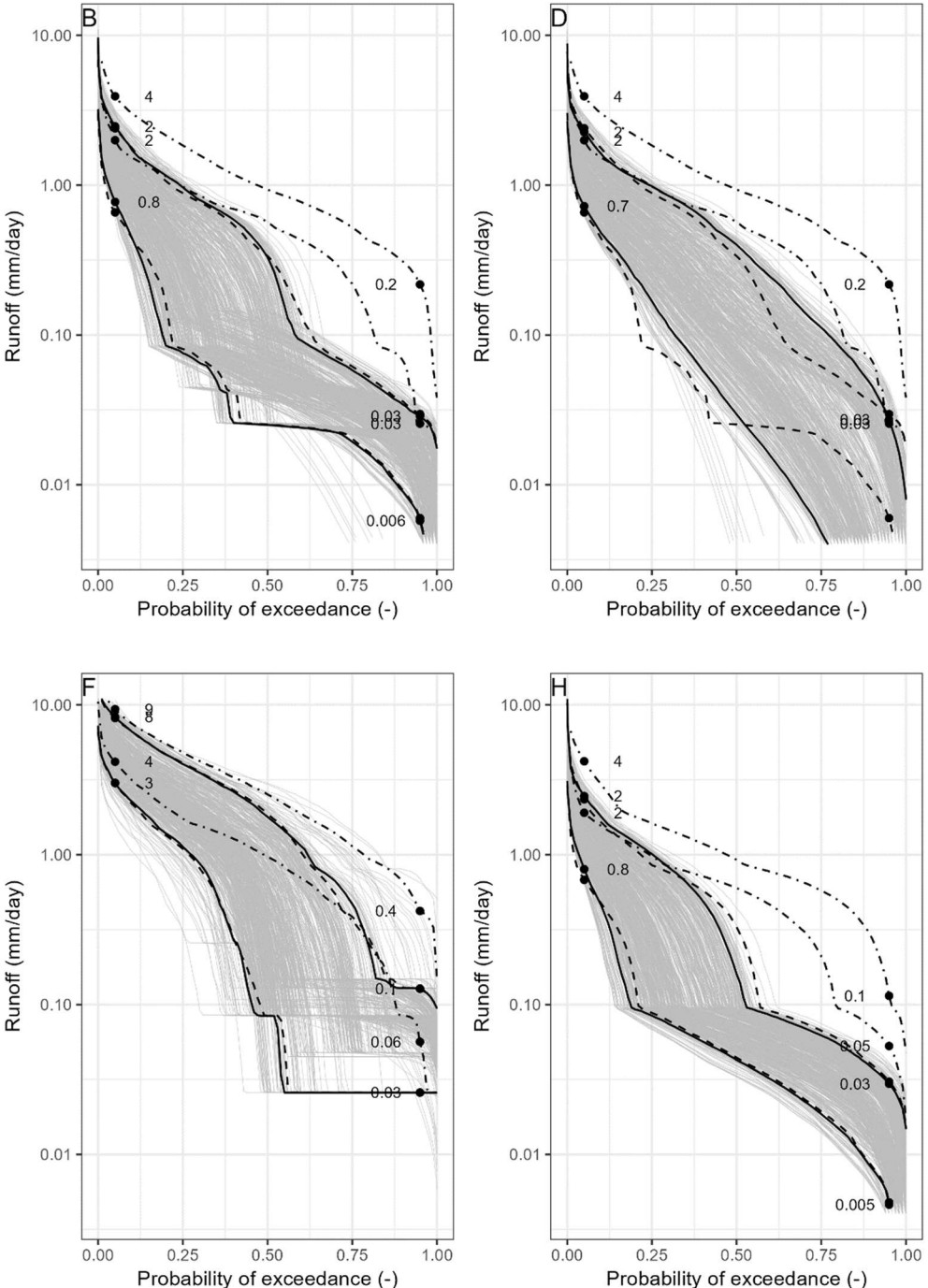

Figure 5: Runoff exceedance curves for coniferous forest on peat (fens with Rewet2). Drained catchments in gray, 5th and 95th percentiles in solid (Drained), dashed (Rewet1) and dot-dashed (Rewet2). R05 and R95 printed for these lines from Drained and Rewet2. n = 656.

Next, we return to the yearly averages of minimum and maximum values, see Fig. 6. The minimum and maximum groundwater level increases up to 0.7 and 0.8 m, but there are also cases and sub-catchments with no increase after rewetting (similar to the results of Holden et al. 2011 and Karimi et al. 2024). This range is a bit larger than the range (95th percentiles) for groundwater level change in the literature review of Bring et al. (2022), which was 0.27-0.63 m increase near the intervention and half as much on average 9 m (range 5-26 m) from the intervention. Their results were not presented in terms of minimum and

maximum yearly values. When comparing Rewet1 and Rewet2, the latter gives substantially larger increases in the minimum and maximum groundwater levels, and the increases are especially large for case D (1.4 m ditches).

The minimum runoff changes between -0.2 mm day$^{-1}$ and +0.5 mm day$^{-1}$. These changes are large when compared with the range in drained minimum runoff presented in Fig. 5. The maximum runoff changes between -1 and +6 mm day$^{-1}$, with at least the upper end being substantial when compared to typical high-runoff values (Fig. 5). Rewet2 gives larger minimum and

maximum runoff compared with Rewet1 as expected. The relationship between groundwater extremes and runoff extremes requires some analysis (next section) because increases in minimum and maximum groundwater levels do not give the same response in runoff between cases.

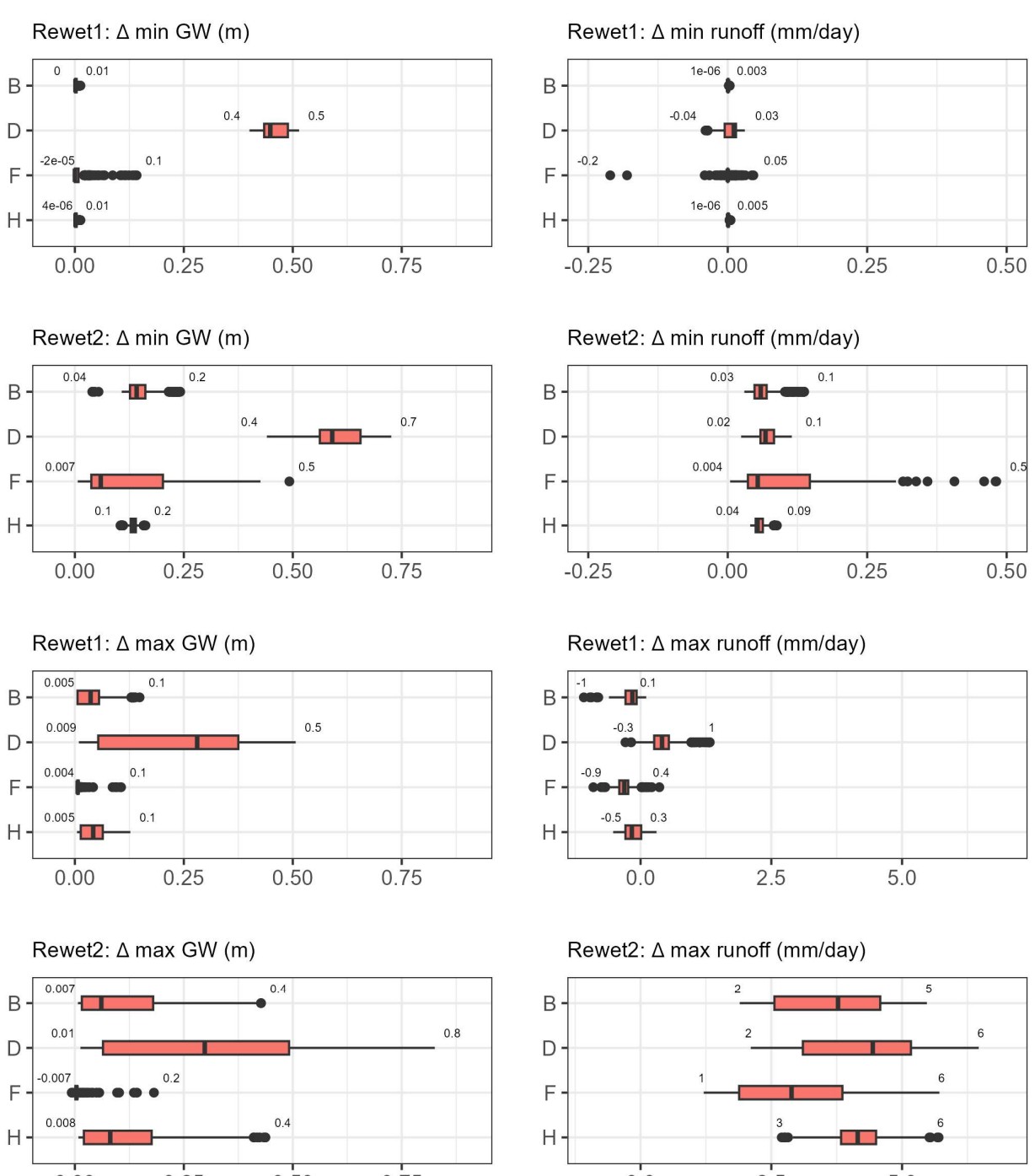

**Figure 6: Change in minimum and maximum groundwater levels and runoff with Rewet1 and Rewet2, for coniferous forest on peatland. n=656 per case (sub-catchments with coniferous forest on peat of depth 1.5 m).**

### 3.3 Driving factors of variable rewetting responses

Here, we evaluate what factors determine the rewetting response in yearly runoff extremes in drained peatlands with coniferous forest. Groundwater extremes are also shown, as an important part of the analysis.

### 3.3.1 Minimum groundwater levels and runoff

Before showing detailed quantitative results regarding changes in minimum yearly values (in Fig. 8), we briefly explain three different situations that occur, see Fig. 7. The minimum yearly groundwater level often increases with rewetting because of the lost ditch drainage at times of the year when the ditch was active (i.e. the groundwater level was above the ditch depth). Higher groundwater levels are associated with increased soil runoff. If the drained minimum level was below the level of ditches (left), then increased soil drainage is the only effect of rewetting on the minimum runoff, which increases. If the drained minimum groundwater level was instead slightly above the level of ditches (center), the minimum runoff also increases, because the increase in soil runoff is large enough to compensate the small loss of ditch drainage. With higher initial groundwater levels (right), the increase in soil runoff can no longer compensate the loss of ditch drainage, which means that the minimum runoff decreases with rewetting, however this is only true for Rewet1. With Rewet2, the additional wetness following reduced interception and evaporation causes increased runoff also here.

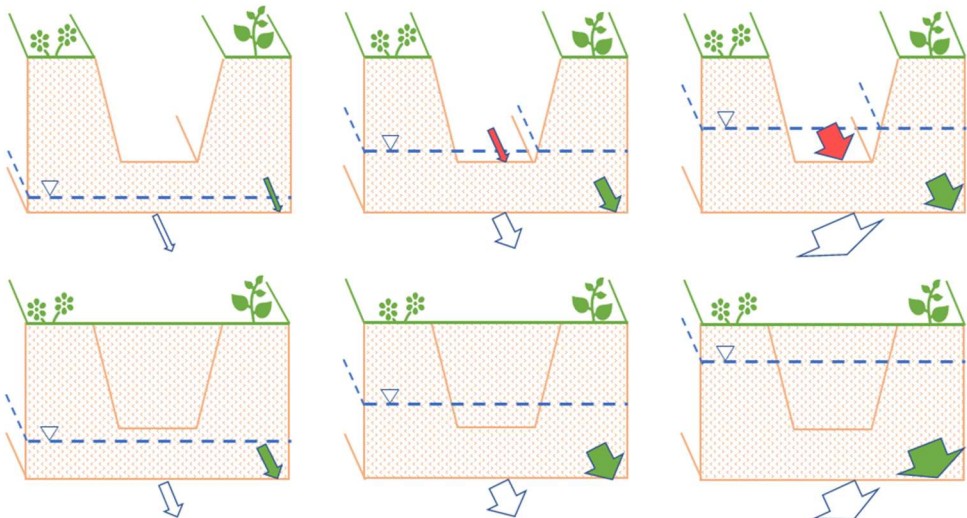

**Figure 7: Schematic of drained (top row) and rewet (bottom row) conditions during the time of the lowest groundwater level per year. Soil runoff (green), ditch runoff (red) and total runoff (white) are represented by arrows.**

Figure 8 shows the quantitative data. Note first the increased minimum runoff for sub-catchments/cases/rewet scenarios to the left of the vertical line, i.e. with drained minimum groundwater levels below ditches (left panel of Fig. 7), although Rewet2 gives much larger increases compared with Rewet1. Even with the wetter conditions of Rewet2, the minimum level often remains in the third soil layer, perhaps because of the higher evaporation losses (which only impact the first and second layer) or higher runoff coefficient above this layer, which effectively remove water from the soil.

Some sub-catchments of case D (1.4 m) and F (recharge) have drained levels that are slightly above the ditch depth (middle panel of Fig. 7), and the small loss of ditch drainage is compensated by increased soil runoff. With higher drained levels, the lost ditch runoff is larger, and with Rewet1 (but not Rewet2), the total runoff decreases.

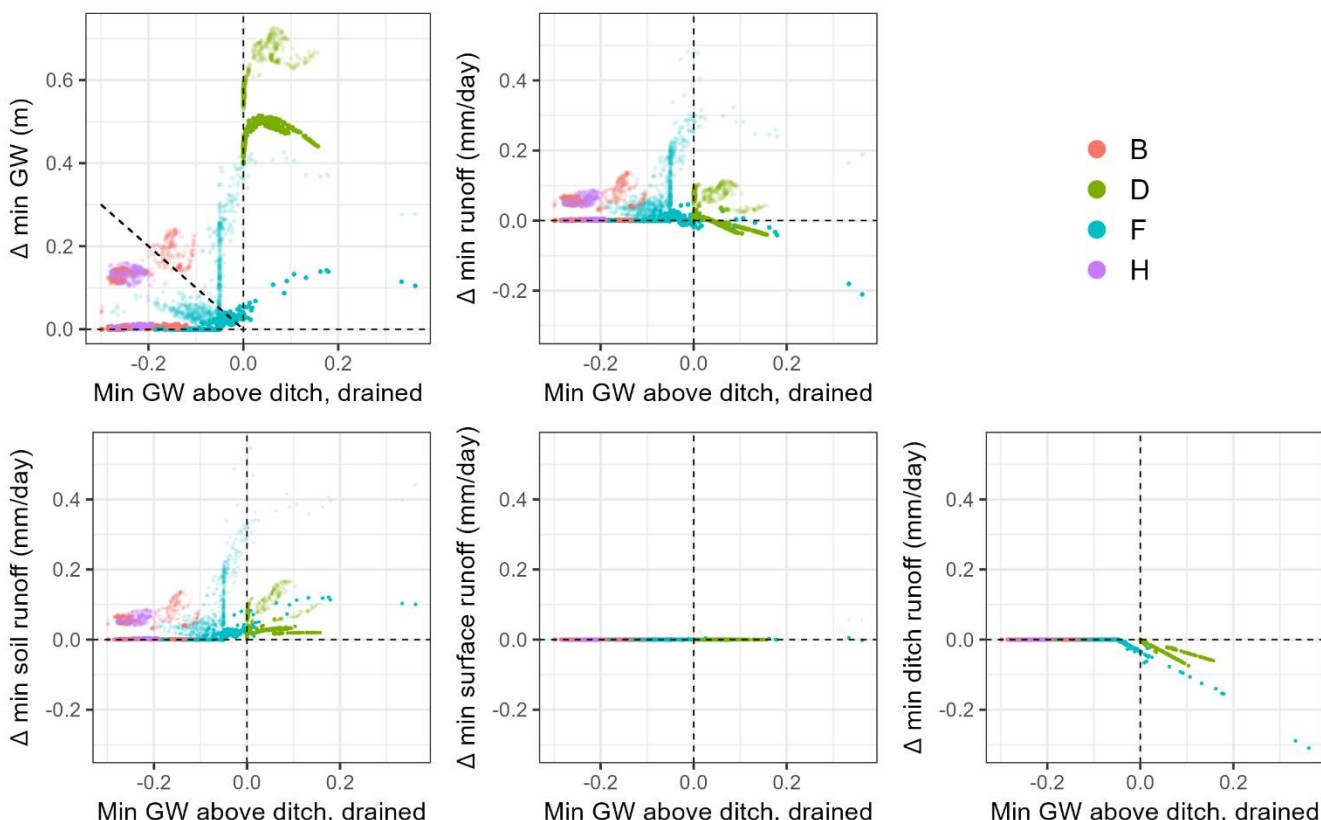

**Figure 8: Changes in average yearly minimum groundwater level and runoff, including the runoff pathways at the time of minimum total runoff, as a function of the drained minimum groundwater level above the level of ditches. Rewet1 cases in full color and Rewet2 cases shaded. At the diagonal line in the top left diagram, a change with rewetting would bring the minimum groundwater level to the level of the removed ditch, which in case B, F and H represents 0.7 m below the surface i.e. close to the lower extent of the second soil layer and in case D it represents 1.4 m below the surface (still in the third soil).**


### 3.3.2 Maximum groundwater levels and runoff

Impacts on the maximum yearly runoff are highly connected with the drained maximum groundwater levels, see Fig. 9. In most cases, the drained maximum level is below the soil surface prior to rewetting (top left panel). If it remains below the surface (left center), the maximum runoff decreases (as desired) because lost drainage is not compensated by soil runoff alone
without the additional "help" from surface runoff. If the level reaches the surface then the total runoff increases instead (lower left) because surface runoff "helps" compensate the lost ditch runoff. If the drained maximum groundwater level was already above the soil surface (top right panel), some cases do not get sufficient increases in the surface runoff to compensate the loss of ditch drainage (right center), meaning that the total runoff is reduced. Other cases get very large increases in surface runoff that cause increases in the total runoff (bottom right panel). Below we explore what causes the difference in behavior between
the center and lower panels.

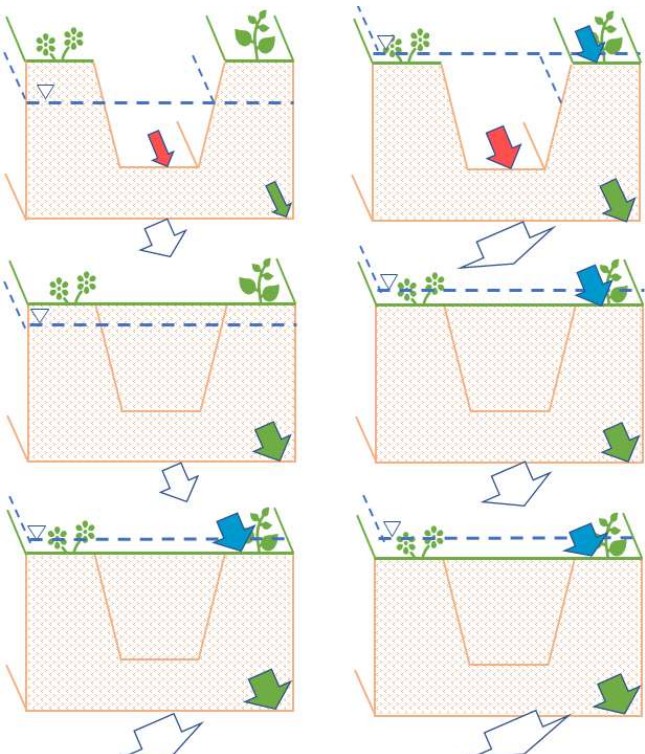

**Figure 9: Schematic of drained (top row) and rewet (middle and lower row) conditions during the time of the maximum runoff per year for situations where the drained level was below (left panels) or above (right panels) the soil surface. Soil runoff (green), ditch**
**runoff (red), surface runoff (blue) and total runoff (white) are represented by arrows.**

Figure 10 shows the quantitative data. Note first the sub-catchments/cases/rewet scenarios to the left of the vertical line, i.e. with drained maximum groundwater levels below the soil surface (left panel of Fig. 9), which is most common. When the maximum level reaches the surface after rewetting, the maximum runoff is increased. This almost always occurs with Rewet2, with large increases in the maximum runoff mostly in the range 3-6 mm day$^{-1}$. With Rewet1, only some (of these originally below-surface) sub-catchments reach the surface and when they do, the increase in runoff is smaller, around 0-1 mm day$^{-1}$, or even with small reductions in some sub-catchments. Here, with lower drained levels, the levels remain below the surface and the runoff is almost unchanged.

With case F (recharge), the drained maximum level was already above the surface due to the additional inflow from recharge areas. Here, with Rewet1, the loss of substantial ditch drainage after rewetting overshadows the increases in soil- and surface runoff, reducing the total runoff by up to 1 mm day$^{-1}$. With Rewet2 (case F), the total runoff increases instead (around 1-5 mm day$^{-1}$), due to larger increase in the surface runoff. Some cases here have a minor decrease in soil runoff despite a small increase in groundwater level, but this is only because they represent different times. (The groundwater level is printed at the end of the time step but does change within the time step, for example with heavy rain and surface runoff, meaning that the day of the maximum total runoff (the day we print soil runoff) can be different from the day of the maximum groundwater level, and even when these days are the same, soil runoff is calculated early in the time step and therefore more affected by the groundwater level from the previous time step. The timing of maximum runoff in case F is impacted by surface runoff which may be why the perceived discrepancy was only seen here.)


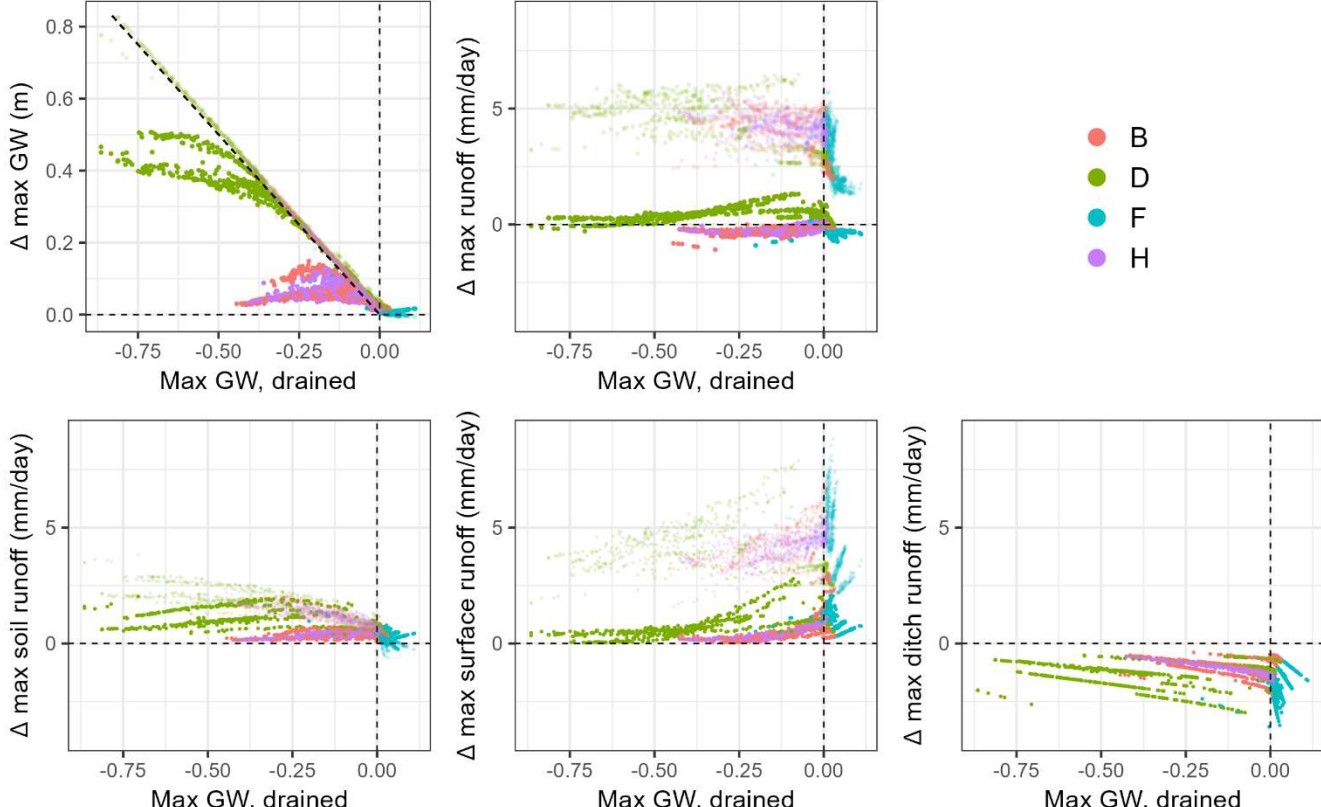

**Figure 10: Changes in average yearly maximum groundwater level and runoff, including the runoff pathways at the time of maximum total runoff, as a function of the drained maximum groundwater level. Rewet1 cases in full color and Rewet2 cases shaded. At the diagonal line in the top leftdiagram, a change with rewetting would bring the maximum groundwater level to the soil surface.**

## 4 Implications for policy makers

The results presented here imply the following related to the potential of rewetting of ditched forested peatland to increase water security in Swedish streams:

Rewetting of these lands unfortunately *cannot* help improve water security (increasing low-flow or reducing peak flows) in catchments of size 10 km² or more. We base this conclusion on the extensive analysis of simulation results where the change in minimum and maximum yearly discharge was less than 5 %. Here, we also remind the reader that the study design already implies an over-estimate of the impact because rewetting was applied to all drained forested peatlands, which is not in the current plans (only around 0.1 million hectares of 0.7 million hectares nationally will be restored). In addition, we assumed perfect recovery to undrained conditions, which would probably not occur, or take a long time. The peatland area considered for rewetting is simply a too small fraction of the surface area of Sweden (1 %), to have any significant buffering impact on water discharge.

The question is then if rewetting can impact extreme flows at smaller scales, which could be important e.g. for local biodiversity in those streams. The largest relative impact would be obtained in small streams draining only peatlands that were fully impacted by drainage before rewetting and then fully restored. For these cases, we can conclude the following regarding the runoff from the drained and rewet peatland:

- Rewetting with restoration to naturally lower tree density (in addition to ditch blocking) often results in substantial increases in simulated low runoff, with up to a factor 10 increase. If tree density was unchanged, changes in low runoff were smaller, and here, very active ditches prior to restoration (deep ditches or wet soil due to lateral inflow), sometimes resulted in reduced low runoff after rewetting. In other words, rewetting *can* help improving water security related to increased low flow in small streams draining only the rewet peatlands, if restored conditions mimic those

of original wetlands, including reduced tree cover.

- Similarly, simulated high runoff in small streams draining only rewet peatlands is only substantially impacted if conditions are restored to the natural conditions of wetlands, but unfortunately substantial changes only occur in the opposite direction to what is desired by water managers, with higher high flows. If the peatland was already wet due to lateral inflow, the changes in high runoff are sometimes smaller. This means that rewetting generally *cannot help*

improving water security related to high flows in these small streams, and that the situation is expected to *worsen* if or when conditions are returned to those of original wetlands.

The analysis of changes in groundwater extremes was only included in this study to understand flow extremes, but we note shortly that the minimum and maximum groundwater levels increased substantially in many cases, and that the range of impact was a bit larger than in a recent literature review by Bring et al. (2022). Changes in groundwater levels might have implications

for other ecosystem services as well as risks, related to e.g. forest fires, greenhouse gas emission, water quality, biodiversity, recreation and pests. This is now a very important and extensive research field in Sweden (e.g. SLU, 2023).

Rewetting with restoration of topographical barriers was not studied here, and might better be described by the literature on constructed wetlands with defined outflow sections.

## 5 Conclusions

From this model experiment, the following conclusions could be drawn regarding rewetting of forested peatlands to solve problems with extreme flows in Sweden and to identify the main causes behind the diverse impacts noted in the literature

- Rewetting drained forested peatlands is not a method that will increase water security related to too little or too much water in the landscape in Sweden (catchments of size 10 km$^2$ or more), as these lands constitute a too small fraction in the landscape (1 %).

- Tree cover is a major driver to discrepancy of hydrological impacts in small streams draining from rewetted peatlands. Hydrological drought can be reduced if trees are removed during the restoration, however, high flows are then likely to increase. To instead reduce high flows the trees should thus be kept. Hence, it is important to decide which of these extremes to address with a specific rewetting activity depending on the conditions downstream.

- Groundwater levels often increase substantially with rewetting, which is the purpose and thus expected. The groundwater levels before rewetting were found to be of major importance for the hydrological impact.

It would be ideal to compare conclusions from this work with field observations. Variable impacts on flow extremes observed in field studies can be easier understood if the following data is recorded:

- Catchments characteristics: area/land use/soils of catchment and area/land use/soil of restored area (in relation to the full catchment)
- Drained conditions: depth of ditches, dynamic groundwater levels including the lateral influence of ditches in transect groundwater wells, extreme groundwater levels in relation to ditch depth and soil surface
- Type of rewetting performed/achieved: ditch blocking performance and change in tree cover density, and impacts on groundwater levels

## Appendix A. HYPE model overview

This Appendix gives a complementary overview of the HYPE model and more details for specific parts important to modeling hydrological effects of peatland restoration.

*Stores and fluxes within and between sub-catchments*

Spatial variability in water balance and fluxes across a landscape is simulated through meteorological forcing, topography, hydrological response units (HRU), human alteration and features of waterbodies along the river network. When setting up the HYPE model, a river-basin or catchment is therefore divided into parts, not overlapping, called sub-catchments, or subbasins, which are defined by topography. At each timestep, forcing through precipitation and temperature initiate alterations of the air-land-vegetation-soil-groundwater processes, which together with human withdrawals, discharges and dam regulations, cause a response in waterbodies in the river network and eventually discharge at the subbasin outlet (Figure A1).

Upstream subbasins contribute to a downstream subbasin with an inflow to the subbasin's main river. The subbasin river network can be composed of the main river, a lake at the outlet of the subbasin, and one conceptual local stream and lake that receive (part of) the runoff from the subbasin's land. The land area of a subbasin is divided into HRU's composed of combinations of land use, soil type and soil depth. The HRU's are normally not influencing each other; each responds to the forcing data (precipitation and air temperature) depending on the characteristics and parameters of the HRU (A and B in figure A1). The surface waters (e.g. lakes, rivers and constructed ponds) depend on inflow from surrounding or upstream areas. They are calculated in series for the river network located within the subbasin (C in figure A1).

The two rewetting scenarios analyzed correspond to the forested peatland HRU in figure A1:B. Scenario Rewet1 (restoration by filling ditches) is represented in the model by removal of pathway runoff through ditch drainage. Scenario Rewet2 (changing land use of the drained part to forested wetland, in addition to filling ditches) is represented in the figure by changed fluxes due to evaporation, interception and surface runoff (size of flows). Note that in figure A1:A the ditch drainage is located in soil layer three similar to the assumptions C and D in the sensitivity analysis of conditions prior to rewetting (Table 3 in the main text), whereas in other simulated assumptions it is located in soil layer two, and the variability was included to analyze

the impact of ditch depth on restoration performance. Assumptions E and F (recharge-discharge area model) correspond to a fraction of the runoff from the soil in figure A1:A being diverted to the soil in figure A1:B, creating wetter conditions there.

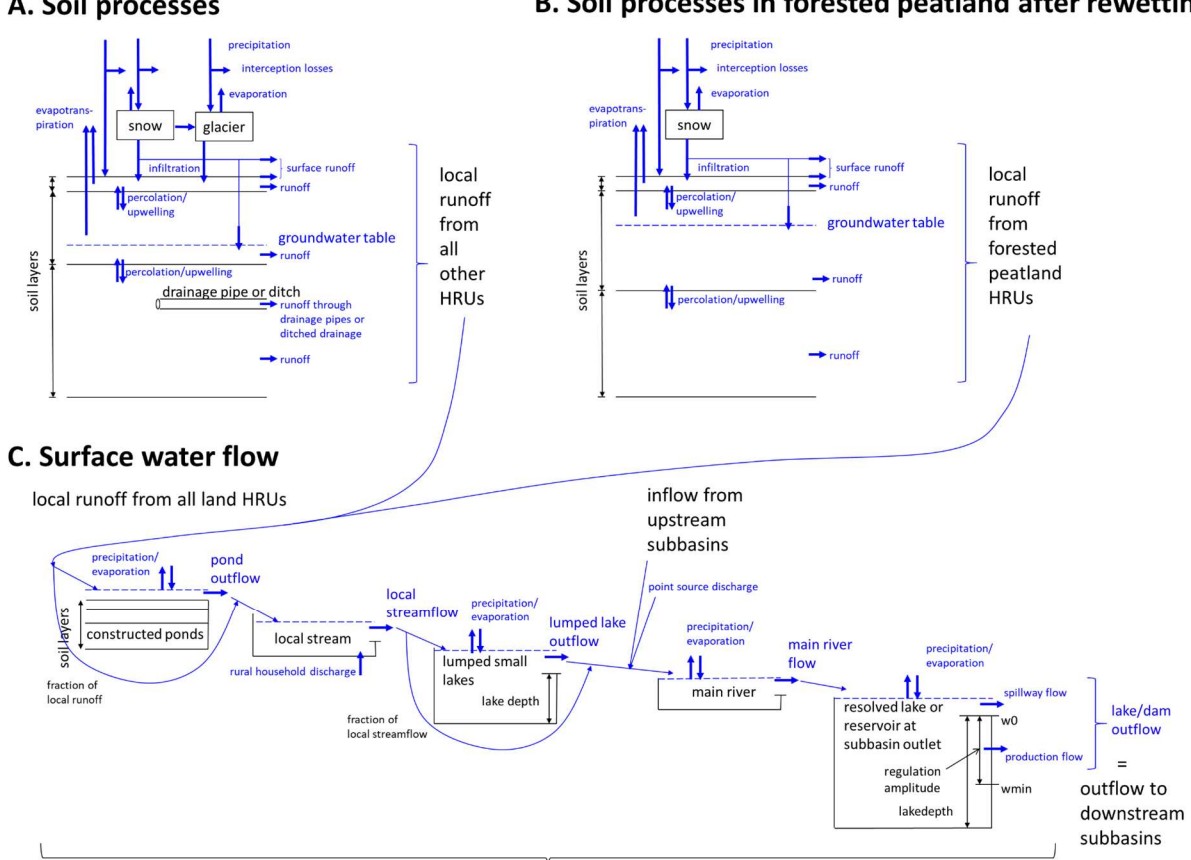

**Figure A1: Overview of water flow, fluxes and storage simulated in a HYPE subbasin, illustrated as vertical profiles of the water balance in different segments of the landscape, where A and B flow into C. Each scheme shows the processes accounted for in: A) each soil/vegetation layer, or snow and ice cover; B) rewet peatlands; and C) potential waterbodies of the surface water network in each subbasin.**

Wetland HRUs separate themselves from other HRUs by having other values of model parameters. In the parameterization used here (version S-HYPE2016i) three HRUs with peat soil are simulated, in addition to one constructed pond HRU. Common differences between wetlands and other HRUs are a lower evapotranspiration rate and larger interception consistent with less vegetation and a higher snow melt rate due to higher insolation (than e.g. forests). They have higher water content than other soils, and lower runoff coefficients than e.g. moraine. One of the wetland HRUs is "forested wetland" (on peat soil). In the study we explore the rewetting of forests on peat, i.e. the soil type does not change with rewetting.

Interception losses are simulated as a reduction of the precipitation before it reaches the soil surface of an HRU. The HYPE parameter in question, *pcluse*, depends on the land use and is defined as the removal fraction of precipitation, i.e. precipitation is multiplied by the factor (1-*pcluse*), see Table 2.

The actual evapotranspiration is calculated as a fraction of a modelled potential evapotranspiration (PET). The model in this paper uses HYPE's default model algorithm for PET together with regional calibration. The algorithm is temperature and land use dependent, and has a sinusoidal seasonal adjustment. The regional correction factor *cevpcorr*, adjusts the rate in larger regions. HYPE does not simulate evaporation separate from transpiration. PET is calculated for the top two layers, assuming an exponential decrease with depth, if the air temperature is above a threshold (parameter *ttmp* in Table 2). In addition to PET, actual evapotranspiration of the land HRUs depends on the current soil moisture relative to field capacity and is linearly reduced to be zero at the wilting point.

### Appendix B. Calibration method and model uncertainty

The S-HYPE model was calibrated in two steps; first finding general parameter values for the whole domain (i.e. the country of Sweden), and thereafter adjusting regional deviations for key parameters from these general values. The general parameters are of three types, 1) constant in the whole model domain, 2) coupled to land use, and 3) coupled to soil type. These parameters were calibrated for 153 small and medium-sized basins (<2000 km²), unaffected by hydropower regulation. The general model parameters were calibrated manually, by compromising between the overall agreement (focusing on the NSE criterion) for the basins with discharge observations, while maintaining the parameters within realistic ranges. Since the S-HYPE model is also used for water quality applications, special attention was paid to the behavior of groundwater levels, surface runoff, water flow paths, and the water holding characteristics of different soil types during calibration. By including these pathways, we were able to represent observed changes in groundwater levels with rewetting (similar range), as shown in the current study. For daily discharge, the mean NSE is 0.86 and volume error -3 % (cf. 0.74 and -2 % in 350 stations that also include those impacted by hydropower). The model performance was further evaluated versus catchment characteristics, to make sure that no trends in volume errors versus e.g. land use remained (see e.g. figure B1).

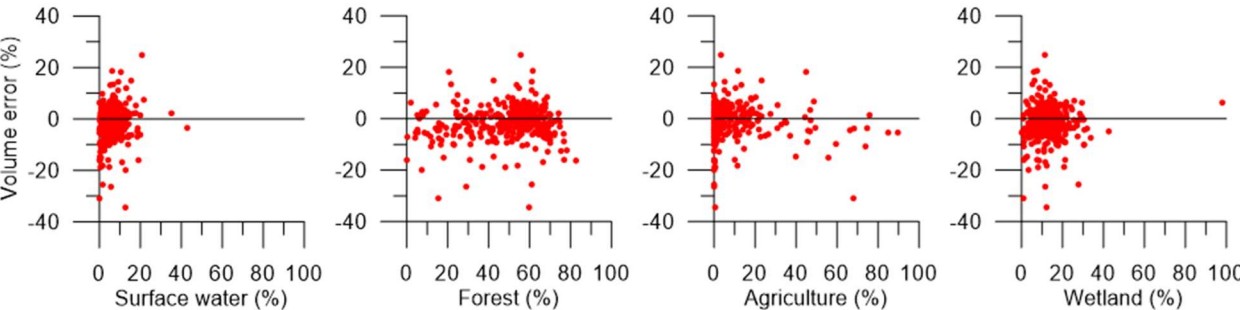

**Figure B1. Scatter plot of volume errors versus percentages of surface water, forest, agriculture and wetland, 350 basins with at least 10 years of observations.**

The regional deviations of key parameters within selected parameter regions govern how these parameters deviate from the general values. They are a compromise between a completely general model and a model where all parameters are calibrated individually for each gauging station. The regional deviations are calibrated based on the regional agreement and the deviation

500    from the general values. The long-term vision is to continuously improve the model structure and input data, thereby reducing the need for regional adjustments. The S-HYPE model is currently used in both versions. The general set-up, without the regional adjustments, for instance for water balance studies, and with the full regional adjustments, for e.g. forecasts and warning services. In this paper both versions are used as the impact of regional calibration is assessed (cases G, H compared with A, B).

505    In this study of impact on extreme flows from rewetting peatlands, model performance on high and low flow are of particular interest. Figure B2 shows an evaluation of low flows (MLQ: mean of the lowest flows for each year), mean flow (MQ: mean flow), and high flows (MHQ: mean of the highest flows for each year). The corresponding NSE for each data set is 0.74, 0.99 and 0.95. The figure illustrates that the relative uncertainty is largest for low flows, with a slight average overestimation, whereas the high flows are often slightly underestimated. The errors in average discharge (MQ) are typically negligible. The

510    discrepancy between modelled and recorded values are, however, not only due to uncertainties in the modelled values. The recorded values, are in fact, not really measurements, but estimated through rating curves, with large uncertainties in the extremes (both low and high), due to real life problems such as damming by for instance ice or vegetation, missing observations, and other perturbations to the river section in question. The data extracted from the database have a resolution of 1 L/s, which is a further indication of the many uncertainties in the recorded data, especially during low flow.

515

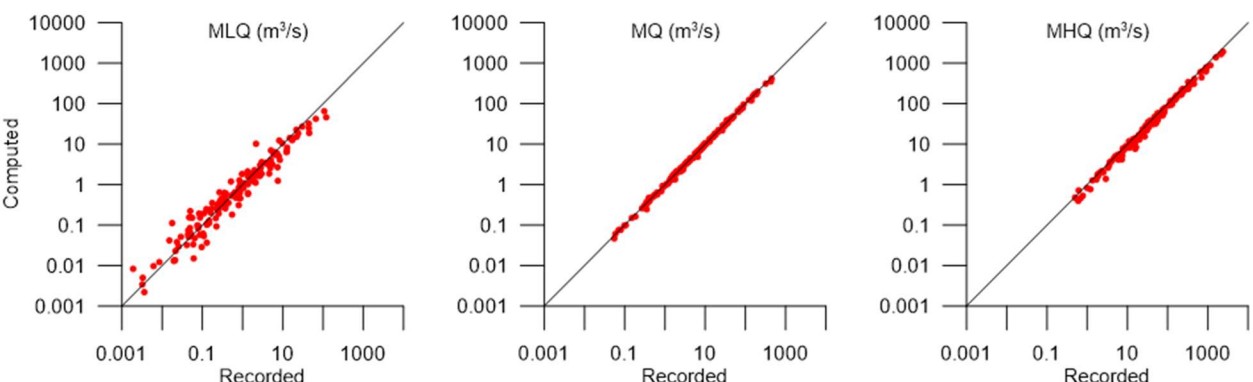

**Figure B2. Evaluation of annual low flows, mean and annual high flows, daily means per station. NSE = 0.74, 0.99 and 0.95 respectively. 157 stations with at least 15 years of complete daily data, with negligible hydropower regulation.**

520    In summary, the model is able to produce realistic responses to both dry and wet conditions, and because it was developed also for water quality assessments, it contains necessary pathways and provides similar local groundwater responses to rewetting

as found in the literature. In addition, the setup of the current study effectively removes uncertainties related to meteorological variability, which often disturb conclusions from experiments using only observed data from field studies. We therefore consider this model to be highly suitable for the analysis of hydrological impacts from rewetting forested peatlands in Sweden.

*Data availability*

The processed simulation results are available at https://doi.org/10.5281/zenodo.13472209. The open-source HYPE code with documentation is available at www.hypeweb.smhi.se. Time series of discharge with S-HYPE version 2016i are available at https://vattenwebb.smhi.se/archive/V-2024-05-21/.

*Author contributions*

ME conceived the study. GL and CP made HYPE code developments. ME and SS performed HYPE simulations and analysis. CP and BA contributed to interpretation of the results. ME and BA designed the manuscript structure. ME and SS wrote the initial draft and ME, CP and BA contributed to the final draft. CP wrote Appendix A and GL wrote Appendix B. ME and BA contributed to the Appendices.

*Competing interests*

The contact author has declared that none of the authors has any competing interests.

*Acknowledgements*

This work was co-financed by the Swedish Environmental Protection Agency (contract No. NV-01874-23 "Support on wetlands SMHI", and research grant NV-08138-18 project "Eviwet") and by the Swedish research council Formas (grant No. FR-2022/0006 project "Fair Water"). The authors would also like to acknowledge help with data processing of ditches by Kristina Isberg at SMHI and constructive feedback from two anonymous reviewers. The study was inspired by the International Association of Hydrological Sciences (IAHS) "Science for Solutions" decade 2023-2032 (HELPING).

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
