# Peer review of "Where can rewetting of forested peatland reduce extreme flows?"

_Hydrology and Earth System Sciences, 2024_

## Author Response (AR1)

Correction of manuscript "Where can rewetting of forested peatland reduce extreme flows? - model experiment on the hydrology of Sweden" *By Elenius et al.* https://doi.org/10.5194/hess-2024-271

**Responses to Referee comments**

*Referee #1:*

**General comments**

Rewetting of peatland does impact greenhouse gas emissions, biodiversity and extreme low and high flows. The authors investigated the effect of rewetting on high and low flows in Sweden. They employ an hydrological model (S-HYPE), which has been calibrated across Sweden. In addition to the national scale, they performed a sensitivity analysis with S-HYPE in a part of Sweden to determine which factors determine the hydrologic effect of rewetting. They show that at a larger scale (> 10km$^2$) rewetting has hardly any impact on extremely high and low flows. At a smaller scale (< 10km$^2$) the effects are mainly determined by the groundwater levels before rewetting and the reduction of tree cover after rewetting.

The current insecurities regarding rewetting policies, extensive distributed modelling of rewetting and the systematic sensitivity analysis make the study very relevant to readers of Hydrology and Earth System Sciences. The modeling study is novel and allows them to answer their research questions. I especially appreciate Figures 7 and 9, which explain clearly why rewetting does not always result in higher low flows or larger high flows.

In general, the paper reads fluently and is well structured. Below I list several specific comments which would strengthen the paper.

**Reply:** Thank you very much for the summary and positive judgement. Please see responses to Specific comments below.

**Specific comments**

Line 39: The term "air-water holding capacity" is not common among hydrologists. Please explain what you mean.

**Reply**: Thank you, we changed to the correct meteorological term: moisture holding capacity of the air.

Line 94: "whereas peat soils cover 17 % of the entire surface and 7 % of forests". Probably you mean that 7% of the entire surface is forested peat soil. This is not clear in your sentence.

**Reply**: Thank you, we clarified in the manuscript which now reads "whereas peat soils cover 17 % of the entire surface and **cover** 7 % of forests"

Line 115: "following 5 years of initialization". Do you mean that the initialization occurred in the years 2007-2011? Clarify in the text.

**Reply**: Yes, the interpretation is correct and we have now clarified this in the manuscript.

Line 119: "these HRUs are described using three soil layers extending to 0.25 m, 0.7 m and 1.5 or 2.25 m below the soil surface". In the next paragraph, you describe the land cover data. Mention also the source of your soil profile data.

**Reply**: Thank you, this was missing. We added and adjusted the text as follows:  "Total soil depth was collected and simplified from Daniels and Thunholm (2014) and calculations are typically performed within three soil layers that extend to depths originally motivated by top soils (affected by plough depth in case of agricultural fields), the remainder of the root zone, and any soil beneath the root zone. The main focus here is on forested peatlands, and the three soil layers of these HRUs extend to 0.25 m, 0.75 m and 1.5 or 2.25 m below the soil surface." (Note, we also corrected a typo mistake for the depth of soil layer two from 0.7 to 0.75 m.) Reference: Daniels, J., and Thunholm, B.: Rikstäckande jorddjupsmodell (National soil depth model). SGU-rapport 2014:14, https://resource.sgu.se/produkter/sgurapp/s1414-rapport.pdf, 2014.

Line 142: Table 2 should be Table 1.

**Reply**: Thank you, very observant. This is updated  now and we checked that the reference to the table in the manuscript is correct.

Line 146: In the baseline scenario you assumed a ditch depth of 0.7 m. As ditch depth is important in the sensitivity analysis, what is the base of 0.7 m?

**Reply**: Thanks, that information was missing. We added reference to Piirainen et al (2017), who provide the depth 0.5-1 m for drainage of ditches in Sweden.

Lines 163-167: You show that the change of tree density after rewetting has a significant impact on interception and evapotranspiration and thus on high and low flows. As your results entirely depend on modeling, we should be sure that sound modeling concepts are used for interception and evapotranspiration. Describe the used concepts for interception and evapotranspiration and show how reliable the concepts are.

**Reply**: We added two new appendices to further describe the model and its reliability. The concepts used for simulating interception and evapotranspiration have been added within a model description in Appendix A. We also show in Appendix B that the volume error is not changed with increased amounts of wetland (i.e. with changed interception and evapotranspiration).

Line 163 and Table 2: You discuss the reduction in tree cover and its effect on model parameters. How large is the reduction in tree cover that you have in mind?

**Reply**: In reality this could vary from site to site, but in the model study we changed from forest to forested wetland where the latter has >10 % tree cover in the NMD dataset (a bit ambiguous as there is no upper limit), meaning all trees were not removed. The model does not explicitly account for tree density, but only models the impact through calibrated parameters depending on HRU. As written in the manuscript, calibrated parameters do not change much between forested wetland and open wetland (no tree cover), as stated in the Table 2 caption.

Figure 2: The amount of information is relatively small in relation to the space it takes. You can explain the main result in the text and, if needed, move the Figure to Supplementary Material.

**Reply**: We agree that the figure does not show much details, but we prefer to keep it in the main article, because it actually contains our main result, showing how little impact there is from rewetting on extreme discharge. We think that core messages should be presented in figures, and even if the message is a lack of change, it is a very important result.

Figure 9: The difference is unclear between the sub-figures right-center, left-bottom and right-bottom. Do you need these 3 subfigures?

**Reply**. We understand the point made here but even though some sub-figures show similar sizes of arrows, they represent different scenarios, as the left panel describes a situation where the drained groundwater level was below the surface and the right panel when it was already above the surface. We think it might be too cumbersome for the reader if we remove some sub-figures and refer to the other panel, so we prefer to keep the figure as it is. However, we extended the figure caption to clearly describe the meaning of the left and right panels.

Table 3 and Figure 3: You apply S-HYPE with (Scen. A and B) and without (Scen. G and H) regional calibration. Figure 3 shows that the impact of regional calibration is relatively limited. Mention this when you discuss Figure 3.

**Reply:** Thank you, we agree, this is an important point to make early. We added the following sentence in relation to the discussion on Figure 3: "The impact of regional calibration on minimum and maximum discharge is negligible."

Lines 364-368: This sentence is too long. Split the sentence.

**Reply**: We agree and split it into three sentences, as follows: "We base this conclusion on the extensive analysis of simulation results where the change in minimum and maximum yearly discharge was less than 5 %. Here, we also remind the reader that the study design already implies an over-estimate of the impact because rewetting was applied to all drained forested peatlands, which is not in the current plans (only around 0.1 million hectares of 0.7 million hectares nationally will be restored). In addition, we assumed perfect recovery to undrained conditions, which would probably not occur, or take a long time."

Line 389: This is the conclusion section. Before you discuss the impact for policymakers and field research, answer the main research questions of your study:

- What are the main drivers behind the heterogenous impacts of rewetting on discharge extremes?
- Where can rewetting of forested peatland reduce extreme flows?

**Reply**: Thank you. In fact these questions were answered under the headline 'Impact for policy makers' of the Conclusion section, which was not ideal. We have now removed the headlines under the Conclusion section and reformulated the text to show more clearly the relation to the research questions.

Line 399: Be more specific on implications for other ecosystem services and risks.
**Reply**: The original text was "Groundwater levels often increase substantially and this might have implications for other ecosystem services as well as risks." We added: ", related to e.g. forest fires, greenhouse gas emission, water quality, biodiversity, recreation and pests and pointed at some ongoing research (SLU 2023).

SLU: Rewetting of drained forest wetlands: strategies for implementation and adaptation to future climate, https://www.slu.se/en/departments/aquatic-sciences-assessment/research/forskningsprojekt/active-research-projects/gh/rewetfor/, accessed on 2024-02-09, last updated 2023-09-23.
* * *
This paper carries out a modelling experiment (using the SMHI HYPE model set up for Sweden: S-HYPE) to assess the effectiveness of forest peatland restoration in buffering low flows and high flows. This is achieved by various semi-distributed modelling scenarios based on new data on the distribution of ditches in Sweden and various assumptions about ditch depth and effectiveness to establish baseline simulations. Adjustments to model parameterisation to mimic the various hydrological process impacts of peatland restoration in relevant spatial domains. The modelling concluded that peatland restoration will not have any significant benefits in terms of mitigating high flows or enhancing low flows. This partly reflects the influence of non-drained areas in larger catchments and the fact that many of the effects of peatland drainage are irreversible, at least on normal policy decision making time scales.

This is important work that is likely to have a large readership. Currently large sums of money are being spent globally on peatland restoration with – often overly optimistic - expectations that benefits will accrue in terms of mitigating hydrological extremes and enhancing carbon sequestration. As the authors correctly argue, modelling experiments like that presented in this paper have an important role here, where much experimental work is very small-scale and can yield contradictory findings from often short-term monitoring. So, I find the work both novel and interesting.

That said, in its current form, it impossible to evaluate the results presented. Consequently, I could not yet support publication in HESS for the reasons presented below.

**Reply:** We appreciate the comments from the reviewer and have added important information on the model and its performance towards data in two new appendices, with corresponding updates to the main text as outlined below. These additional items resulted in addition of another co-author, Göran Lindström, who is responsible for managing the S-HYPE model at SMHI.

**Lack of information about HYPE** and its use in this application: The paper provides very little information on the HYPE model. I am aware that it is a well-known model, but even as someone with extensive experience in hydrological modelling, I had to read the original Lindstrom et al. (2010) paper to understand the model sufficiently well to have a basic appreciation of how it functions and how it has been adapted for this application. Many interested readers of the paper will likely be experimentalists who have very limited modelling knowledge. Clearly the paper should be sufficiently "stand-alone" that other papers don't have to be read to have a basic understanding of the work. I strongly recommend as a minimum a basic description of the model work flow and a conceptual diagram of the model and its basic parameterisation (in terms of stores and fluxes) showing how it was adapted to mimic the effects of drainage/peatland restoration. Otherwise, it is impossible for anyone other than an experienced HYPE user to know what has been done.

**Reply**: We thank the reviewer for the time spent to understand the model and appreciate the comment that this should be made easier for the reader. We now added Appendix A devoted to a HYPE model overview with focus on aspects relevant here.

**No information on model calibration**: Virtually no information is given in the text on how the model was calibrated or what the associated uncertainties in predictions are. We are told that the S-HYPE model was calibrated (though the reader is not told how – just referred to another paper pre-dating the calibration period) that for 2006-2020 with a mean NSE for Q of 0.79 and a small volume error. No information is given in the text on efficiency of low flow predictions (despite them being central to

the objectives of the study). Many will see this as being a critical flaw in the work as we have no means of evaluating whether the model is fit for purpose or not.

**Reply**: We thank the reviewer for this comment, which has contributed to the solidity of the paper. It is a recognised problem with modelling studies that the models are often too complex to be described in a short scientific paper and thus demand reading more literature - we have tried to solve this by adding appendices to the paper and a key reference to the on-line model documentation at https://hypeweb.smhi.se/. We have added information on the model concept in Appendix A and the calibration methodology and performance in Appendix B. In short, the model is able to produce realistic responses to both dry and wet conditions, and because it was developed also for water quality assessments, it contains necessary pathways and provides similar local groundwater responses to rewetting as found in the literature. In addition, the setup of the current study effectively removes uncertainties related to meteorological variability, which often disturb conclusions from experiments using only observed data from field studies. We therefore consider this model to be highly suitable for the analysis of hydrological impacts from rewetting forested peatlands in Sweden.

**No evaluation of model uncertainty**: Given the inevitable uncertainties in the modelling: a mean NSE for Q of 0.79 sounds quite good but is suggesting high uncertainties at least for some sites for high flow predictions alone. Consequently, many of the very small percentage changes simulated for the various scenarios in Table 3 are likely within the uncertainty of the model and may not be significant. The authors need to provide more of the evidence they have on why they think the modelling overcomes these issues. Certainly statements like "Rewetting…..cannot help improve water security… …in catchments >10km2" (L364) are strong and unambiguous, yet the reader has no means of assessing their validity on the basis of the information presented in the paper. This also seems potentially dangerous given the policy relevance of such findings.

**Reply**: The main conclusion of small impact in catchments > 10 km2 is mainly because these lands constitute a too small fraction of the landscape (1 %). We now emphasized this further in sections 4 and 5. The new Appendix B provides details on the agreement between the model and observations.

Setting aside these more fundamental criticisms about the presentation of the modelling itself, the paper is quite well-written and the quality of the graphics is good. I am also supportive of the general approach; though I think the title should make clear that this is a modelling experiment. As it stands, many are likely to expect a more empirical data-driven study. Related to this, the authors should remain aware throughout that these are modelling results – not data! Despite the preceding comments, I would be inclined to agree with the study's conclusions. However, because it is such an important near-policy topic, the modelling really needs to be presented in a much more robust way and show more critical thinking in terms of interpreting the results with model uncertainties in mind.

I therefore hope in revision that the authors provide the additional information requested and present their important work in a more robust way that will increase its impact.

**Reply**: Thank you for this suggestion to increase impact. We changed the title to 'Where can rewetting of forested peatland reduce extreme flows? - model experiment on the hydrology of Sweden' and added the appendices as described above. We feel confident with our main conclusions, and think that with the new title and appendices, the reader will be fully aware that it is a modeling study. To further emphasize the importance of field work, we now state in the Conclusion that it would be ideal to compare conclusions from this work with field observations.